Original research

# How prevalent is COVID-19 vaccine hesitancy in low-income and middle-income countries and what are the key drivers of hesitancy? Results from 53 countries

Julia Dayton Eberwein ![ORCID] , Ifeanyi Nzegwu Edochie, David Newhouse, Alexandru Cojocaru, Gildas Deudibe Bopahbe, Jakub Jan Kakietek, Yeon Soo Kim, Jose Montes

World Bank Group, Washington, District of Columbia, USA

**Correspondence to**
Dr Julia Dayton Eberwein; jdayton@worldbank.org

## ABSTRACT

**Objectives** This study aims to estimate the levels of COVID-19 vaccine hesitancy in 53 low-income and middle-income countries, differences across population groups in hesitancy, and self-reported reasons for being hesitant to take the COVID-19 vaccine.

**Methods** This paper presents new evidence on levels and trends of vaccine hesitancy in low-income and middle-income countries based on harmonised high-frequency phone surveys from more than 120 000 respondents in 53 low-income and middle-income countries collected between October 2020 and August 2021. These countries represent a combined 53% of the population of low-income and middle-income countries excluding India and China.

**Results** On average across countries, one in five adults reported being hesitant to take the COVID-19 vaccine, with the most cited reasons for hesitancy being concerns about the safety of the vaccine, followed by concerns about its efficacy. Between late 2020 and the first half of 2021, there tended to be little change in hesitancy rates in 11 of the 14 countries with available data, while hesitancy increased in Iraq, Malawi and Uzbekistan. COVID-19 vaccine hesitancy was higher among female, younger adults and less educated respondents, after controlling for selected observable characteristics.

**Conclusions** Country estimates of vaccine hesitancy from the high-frequency phone surveys are correlated with but lower than those from earlier studies, which often relied on less representative survey samples. The results suggest that vaccine hesitancy in low-income and middle-income countries, while less prevalent than previously thought, will be an important and enduring obstacle to recovery from the pandemic.

## INTRODUCTION

The world is entering the third year of the global COVID-19 pandemic, which has caused enormous devastation to both people's health (450 million cases and 6 million deaths as of March 2022) and national economies,

## STRENGTHS AND LIMITATIONS OF THIS STUDY

⇒ The greatest strength of the study is that it provides nationally representative and comparable estimates of COVID-19 vaccine hesitancy for a very large sample (over 50) of low-income and middle-income countries, filling a critical evidence gap.

⇒ The study is the first to explore correlates and self-reported reasons for COVID-19 vaccine hesitancy in such a large and comparable sample of low-income and middle-income countries.

⇒ One potential limitation of this study is that it is based on phone survey data which exclude respondents who did not have access to a phone, although access to mobile phones was high in the countries in our sample and where possible sampling weights were used to correct for this bias.

⇒ Another limitation is that the majority of the countries are located in two regions, Latin America and the Caribbean and sub-Saharan Africa, making results less representative of other regions.

in the form of a global recession that has pushed millions into poverty.[1 2] With the continued emergence of new variants and limited treatments available, it is commonly accepted that widespread vaccination is the world's best bet to contain the virus and it is also expected to play an important role in economic recovery.[3 4] As of early March 2022, over 60% of the world's population had received at least one dose of a COVID-19 vaccine. However, there are stark disparities in vaccination rates across countries: only 14% of people in low-income countries had received at least one dose as of 22 March 2022, compared with 79% in high-income countries and 81% in upper-middl-income countries[5] (online supplemental figures S1 and S2). The lag in vaccine distribution in

lower-income countries had until recently put the focus mainly on supply-side constraints. However, with vaccine production now exceeding demand, it is increasingly important to understand the extent of vaccine hesitancy and the nature of the concerns.

Vaccine hesitancy refers to the delay in acceptance or refusal to vaccinate despite availability of vaccination services.[6] It is not a new phenomenon, and some level of hesitancy exists for most vaccines; even before the COVID-19 pandemic began, the WHO identified vaccine hesitancy as one of the top threats to global health.[7] Brewer *et al* established a conceptual framework of behavioural and social drivers (BeSD) of vaccination that has been widely adopted, including by the WHO.[8] These drivers of vaccination can be grouped and measured in four domains: thinking and feeling about vaccines (including confidence in vaccine effectiveness and concern about safety); social processes that drive or inhibit vaccination; motivation (or hesitancy) to seek vaccination and practical issues involved in seeking and receiving vaccination. This framework is focused on proximal factors affecting decisions to vaccinate, which are thought to be potentially changeable by programmes. Several aspects of the thinking and feeling domain are thought to contribute substantially to COVID-19 vaccine hesitancy: the accelerated speed of development of the COVID-19 vaccine, its newness, concerns about longer-term side effects, uncertainty about the duration of the vaccine's effectiveness, and the type of vaccine (eg, mRNA) and country of origin of the vaccine.[9–12]

The multicountry studies that exist show relatively high levels of hesitancy to take the COVID-19 vaccine, but mainly focus on high-income countries.[13 14] Less is understood about COVID-19 vaccine hesitancy in low-income and middle-income countries. A few studies reported multicountry survey results from low-income and middle-income countries[15–20] or systematic reviews of individual country surveys,[21] but few of the studies were based on nationally representative samples of respondents or included comparable results for many countries. Given this limitation, an important contribution of this analysis will be to validate these findings.

This paper extends the existing literature by providing estimates of and reasons for vaccine hesitancy for 53 low-income and middle-income countries, which are comparable across countries and largely based on household surveys that are more representative of the national population. The 53 countries represent approximately 30% of the population of all low-income and middle-income countries and 53% excluding India and China.

## DATA AND METHODS
### Data
This study describes the levels of vaccine hesitancy and its reasons in 53 low-income and middle-income countries between October 2020 and August 2021 using data from the high-frequency phone surveys (HFPS) implemented to monitor the impact of COVID-19.[22 23] Data are available for one survey round from 39 countries and 2 or more survey rounds for 14 countries (online supplemental table S1). The sampling frame was drawn from pre-existing nationally representative household surveys in 19 countries, random digit dialling (RDD) in 29 countries, and a list of phone numbers typically obtained from mobile phone operators in 5 countries (see online supplemental file 1 for detailed description of methods).

### Outcome measures
The question about vaccine hesitancy, 'When a vaccine to prevent COVID-19 is available to you, are you planning to be vaccinated?' varied slightly across surveys depending on whether a vaccine was available in the country at the time of the survey (see online supplemental file 1). In the sample, 47 countries offered three answer options (yes, not sure and no), while the other six offered two categories (yes and no). We combine the 'no' and 'not sure' answers to obtain the measure of vaccine hesitancy.

To obtain the respondent's reason for vaccine hesitancy, survey respondents who answered 'no' or 'not sure' were asked 'What is your (main) reason/concern for not wanting to be vaccinated/not being sure if you want to be vaccinated?' The answer categories varied widely across surveys (online supplemental table S2A). To make these more comparable across countries, answers were remapped into the nine most common categories (online supplemental table S2B). The surveys also differed across countries in terms of whether a single concern or multiple concerns were collected, and to account for this, the results are presented separately.

### Contextual data
Country-level contextual data were drawn from other sources: new COVID-19 cases per million, measured as a 7-day rolling average prior to the midpoint of the month prior to survey data collection in each country[1]; and the Oxford Stringency Index, ranging from 0 to 100, that indicates the degree to which restrictions were put in place by governments to control the pandemic[24]; confidence in the press and government[25] and estimated excess deaths due to COVID-19.[26]

### Analytical strategy
Data from 53 countries were pooled into a single data set. Sampling weights were scaled such that countries are weighted equally; in other words, the estimates of vaccine hesitancy are unweighted averages of population-weighted country averages. (Using population weights when aggregating across countries would have resulted in the results being driven by a small number of large countries.) First, we report point estimates and 95% CIs for each country based on SEs clustered at the state or province level within each country. We also report the simple average across all countries and stratified by: World Bank region; World Bank country income group; urban versus rural residence; gender of respondent; whether respondent is head of household;

age of the respondent; and educational attainment of the respondent. Tukey's test of multiple comparisons was used to test differences in rates of vaccine hesitancy across groups. Second, we used multivariate regression analysis to assess the relative association of vaccine hesitancy and the correlates and contextual variables. Trends over time in levels of vaccine hesitancy were examined in countries with more than one wave of results. Finally, the analysis describes the reasons for vaccine hesitancy. Descriptive statistics were produced using R Software and regressions were estimated using STATA 17.

### Patient and public involvement

No patients were involved in this study. The results of the study will be disseminated to key policy-makers and relevant stakeholders involved in COVID-19 vaccine deployment.

### RESULTS

#### Levels of vaccine hesitancy across 53 countries

Overall, the average level of vaccine hesitancy across the most recent survey round in 53 countries was 20.0% (95% CI 17.2% to 22.7%). Across regions, average levels of hesitancy were highest in Europe and Central Asia (58.8%, 95% CI 55.0% to 62.6%), followed by the Middle East and North Africa (47.4%, 95% CI 38.8% to 56.0%), East Asia and Pacific (26.2%, 95% CI 21.4% to 31.0%), sub-Saharan Africa (15.5%, 95% CI 11.8% to 19.2%), and Latin America and the Caribbean (8.0%, 95% CI 6.5% to 9.5%) (figure 1 and online supplemental table S3). Differences between all regions were statistically significant at the 5% level (online supplemental table S4). When considered by country income group, the highest average level of hesitancy was in lower-middle-income countries (27.7%, 95% CI 23.8% to 31.7%), followed by low-income countries (14.6%, 95% CI 7.8% to 21.4%), upper-middle-income countries (12.7%, 95% CI 9.8% to 15.6%), and high-income countries (5.9%, 95% CI 3.4% to 8.4%). Differences in hesitancy across country income groups were significant at the 5% level across all pairs except for the comparison between low-income and upper-middle-income groups.

#### Bivariate correlates of vaccine hesitancy

Rural households reported significantly higher levels of vaccine hesitancy than urban households (23.2% vs 17.7%, p=0.018), and female respondents (22.5%) were significantly more likely than male respondents (17.3%)

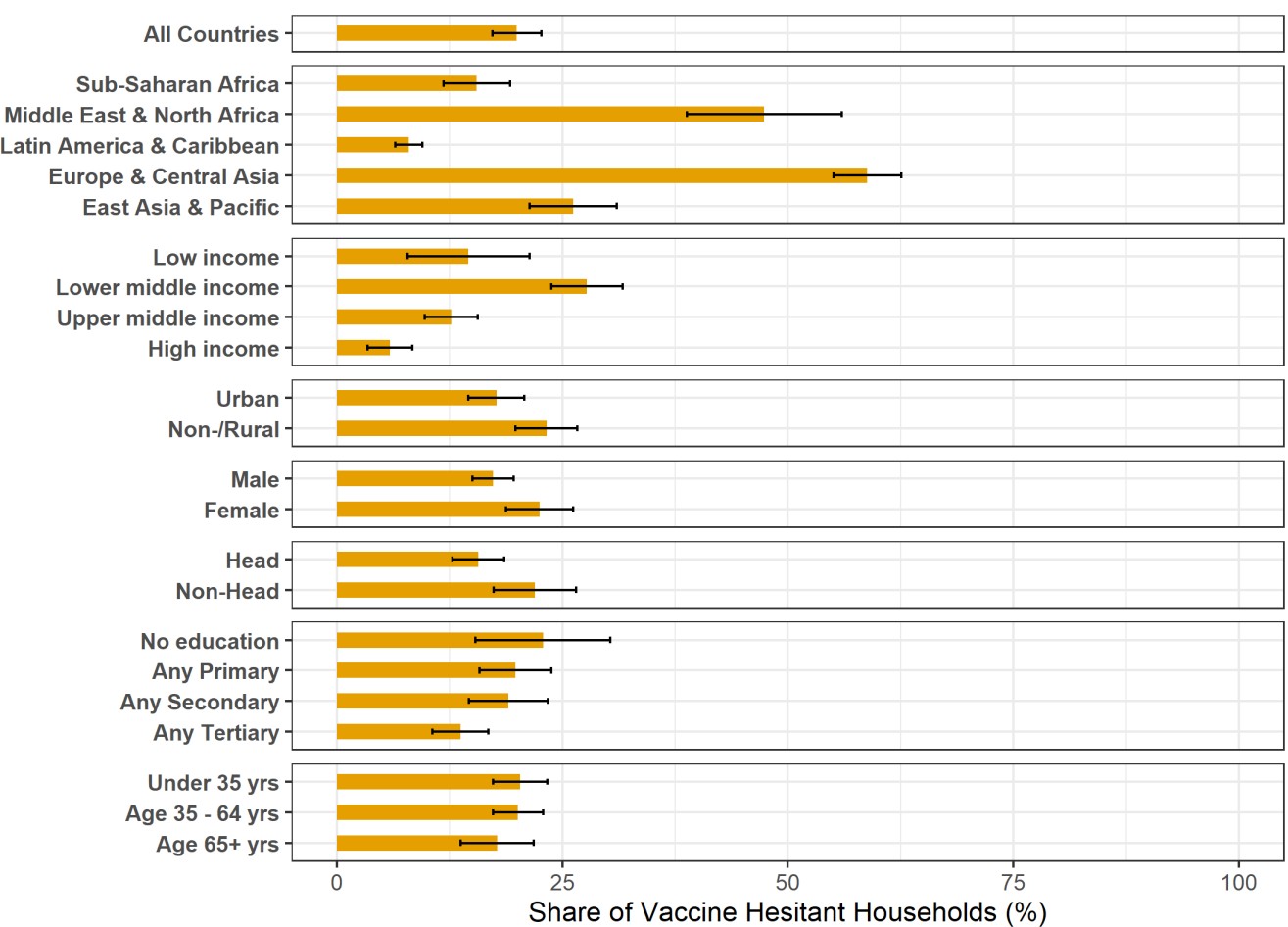

**Figure 1** Share of households that were hesitant to be vaccinated against COVID-19 in 53 countries. Notes: Weighted estimates, with weights scales such that 53 countries are given equal weights, except for results by education which include only results from the 33 country surveys with information on the educational attainment of the respondent.

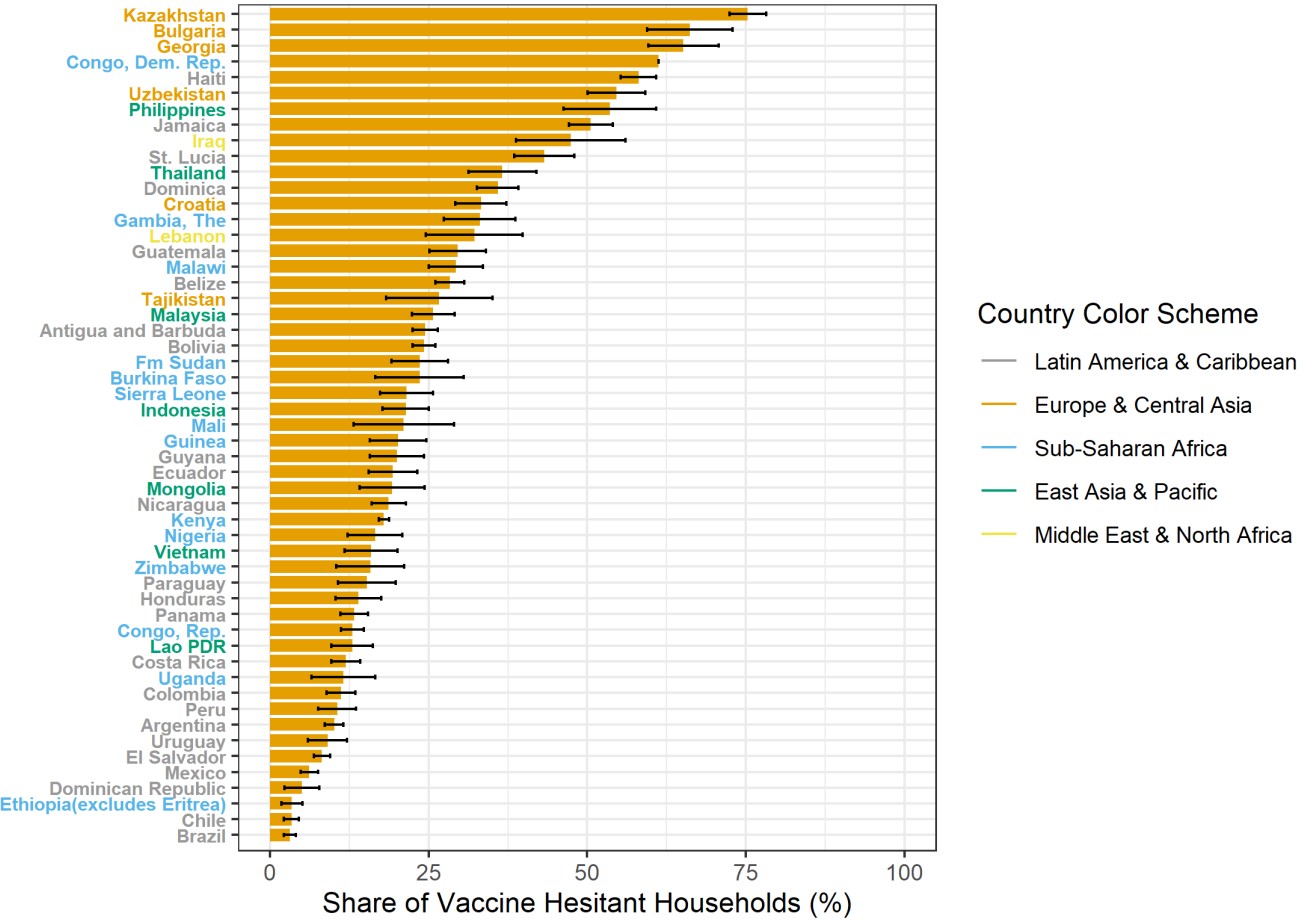

**Figure 2** Share of households that were hesitant to be vaccinated against COVID-19 (answering either 'no' or 'not sure'), by country. Notes: Weighted estimates, with weights scales such that 53 countries are given equal weights. Black bars indicate confidence intervals. Point estimates are given in online supplemental table S3. Source: authors' calculations.

to be vaccine hesitant (p=0.021). Among households in the 33 surveys with information on the educational attainment of the respondent, respondents with lower levels of education had on average higher levels of hesitancy. Younger respondents reported higher levels of hesitancy than older respondents, but the differences were not significant.

## Vaccine hesitancy by country

The highest levels of vaccine hesitancy were reported in Kazakhstan (75.3%), Bulgaria (66.2%) and Georgia (65.2%), and the lowest levels in Chile (3.4%), Ethiopia (3.4%) and Brazil (3.1%) (figure 2). Strength of vaccine hesitancy was available for the 46 countries that included three answer options (yes, not sure and no). In four countries (Philippines, Georgia, Jamaica and Kazakhstan), over 20% of respondents reported being unsure of whether they would get vaccinated, and in another 11 countries, between 10% and 20% of the sample was unsure whether they would be willing to be vaccinated against COVID-19 (online supplemental figure S3).

## Multivariate analysis of correlates of vaccine hesitancy

Results from a multivariate regression generally confirmed the bivariate results. Table 1 shows ordinary least squares

regression results: a baseline model with the correlates described above plus month of survey (expressed in terciles), (time of survey was split into the following three groups: (1) November 2020–January 2021; (2) March–May 2021 and (3) June–August 2021. There are no observations in the data for February and April of 2021.) and extended models with additional contextual variables (expressed in terciles). Results were consistent across models, as were the marginal effects from logit regressions (online supplemental table S5A). Men were less COVID-19 vaccine hesitant than women, although the magnitude was small. Having more formal education was associated with being less hesitant and the differences were more substantial. Age was inversely correlated with hesitancy. There were no statistically significant differences between hesitancy among rural and urban respondents. Differences across regions were statistically significant and, for some regions, quite large, but there were no significant differences by country income group.

In the extended regression models, we tested two additional hypotheses. First, we expected that the severity of the pandemic (for which we use three different proxies: number of new cases of COVID-19 in the country in column 2, the stringency of policy measures as captured by

**Table 1** Multivariate analysis of vaccine hesitancy in 53 low-income and middle-income countries (LMICs) (ordinary least squares)

| Dependent variable: hesitancy | (1) | (2) | (3) | (4) | (5) | (6) |
|---|---|---|---|---|---|---|
| Male | 0.028* (0.013) | 0.027* (0.013) | 0.028 (0.014) | 0.027* (0.013) | 0.031* (0.013) | 0.038** (0.012) |
| Head of HH | 0.012 (0.015) | 0.009 (0.014) | 0.019 (0.013) | 0.000 (0.017) | 0.011 (0.015) | 0.007 (0.013) |
| Education of respondent (ref—no education) | | | | | | |
| Any primary | 0.062 (0.031) | 0.062* (0.031) | 0.056 (0.030) | 0.067* (0.032) | 0.073* (0.034) | 0.073* (0.032) |
| Any secondary | 0.059 (0.035) | 0.060 (0.036) | 0.053 (0.032) | 0.069 (0.037) | 0.067 (0.039) | 0.082* (0.035) |
| Any tertiary | 0.123*** (0.033) | 0.125*** (0.035) | 0.116*** (0.032) | 0.134*** (0.037) | 0.125** (0.040) | 0.143*** (0.036) |
| Age group (ref.—34 and younger) | | | | | | |
| Working age (35–64) | 0.047** (0.014) | 0.048*** (0.013) | 0.042** (0.015) | 0.050*** (0.013) | 0.045** (0.015) | 0.049*** (0.012) |
| Retirement age (65+) | 0.103*** (0.023) | 0.102*** (0.024) | 0.096*** (0.023) | 0.103*** (0.022) | 0.099*** (0.024) | 0.099*** (0.023) |
| Rural area | 0.016 (0.019) | 0.014 (0.018) | 0.013 (0.018) | 0.010 (0.015) | 0.012 (0.015) | 0.011 (0.011) |
| Region (ref.—LAC) | | | | | | |
| EAP | 0.211** (0.076) | 0.218* (0.086) | 0.241** (0.084) | 0.127 (0.076) | 0.299 (0.155) | 0.249 (0.137) |
| ECA | 0.363*** (0.075) | 0.364*** (0.076) | 0.347*** (0.073) | 0.376*** (0.076) | 0.369*** (0.056) | 0.345*** (0.068) |
| MNA | 0.231*** (0.043) | 0.230*** (0.043) | 0.262*** (0.062) | 0.240*** (0.036) | 0.288*** (0.059) | 0.393*** (0.050) |
| SSA | 0.082 (0.075) | 0.089 (0.079) | 0.078 (0.074) | 0.005 (0.101) | 0.072 (0.086) | 0.057 (0.103) |
| Country income group (ref.—LIC) | | | | | | |
| LMIC | 0.073 (0.127) | 0.071 (0.124) | 0.052 (0.122) | 0.090 (0.129) | 0.049 (0.113) | 0.053 (0.081) |
| UMIC | 0.052 (0.154) | 0.050 (0.154) | 0.071 (0.146) | 0.096 (0.154) | 0.026 (0.135) | 0.130 (0.107) |
| HIC | 0.161 (0.159) | 0.160 (0.162) | 0.183 (0.153) | 0.208 (0.162) | 0.175 (0.136) | 0.280* (0.108) |
| Survey month (ref.—November 2020–January 2021) | | | | | | |
| March–May 2021 | 0.073 (0.095) | 0.062 (0.108) | 0.102 (0.104) | 0.056 (0.101) | 0.113 (0.112) | 0.110 (0.095) |
| June–August 2021 | 0.018 (0.090) | 0.012 (0.099) | 0.018 (0.099) | 0.028 (0.089) | 0.049 (0.097) | 0.026 (0.079) |
| New COVID-19 cases per million, terciles (ref.—bottom tercile) | | | | | | |
| Cases (middle tercile) | | 0.022 (0.065) | | | | 0.023 (0.057) |
| Cases (top tercile) | | 0.009 (0.054) | | | | 0.011 (0.049) |
| Oxford stringency index terciles (ref.—bottom tercile) | | | | | | |
| Stringency (middle tercile) | | | 0.061 (0.073) | | | 0.148 (0.077) |
| Stringency (top tercile) | | | 0.005 (0.060) | | | 0.028 (0.059) |
| WHO excess deaths due to COVID-19 (ref.—bottom tercile) | | | | | | |
| Excess deaths (middle tercile) | | | | 0.110* (0.050) | | 0.123 (0.080) |
| Excess deaths (top tercile) | | | | 0.059 (0.067) | | 0.045 (0.106) |
| Confidence in government index tercile (ref.—top tercile) | | | | | | |
| Confidence in government (middle tercile) | | | | | 0.073 (0.054) | 0.105 (0.064) |
| Confidence in government (top tercile) | | | | | 0.016 (0.131) | 0.055 (0.135) |
| Constant | 0.405** (0.138) | 0.411** (0.143) | 0.381* (0.146) | 0.529*** (0.142) | 0.331* (0.157) | 0.333* (0.153) |
| $R^2$ | 0.090 | 0.090 | 0.095 | 0.097 | 0.102 | 0.123 |
| N | 65 088 | 65 088 | 65 088 | 65 088 | 65 088 | 65 088 |

Weighted OLS regressions. Marginal effects reported. SEs clustered at country level.
*, **, *** indicates significance at the 95%, 99% and 99.9% level.
EAP, East Asia and the Pacific; ECA, Europe and Central Asia; HH, household; HIC, high-income country; LAC, Latin America and the Caribbean; LIC, low-income countries; MNA, Middle East and North Africa; OLS, ordinary least square; Ref, reference group; SSA, sub-Saharan Africa; UMIC, upper-middle-income countries.

the Oxford Stringency index in column 3, and estimated COVID-19 related excess deaths in column 4) would be negatively correlated with hesitancy. In our data, none of the three proxies for pandemic severity was statistically associated with vaccine hesitancy except for COVID-19 excess deaths, where in countries in the second tercile of the excess deaths distribution vaccine hesitancy was, on average, lower, relative to the bottom tercile baseline. We

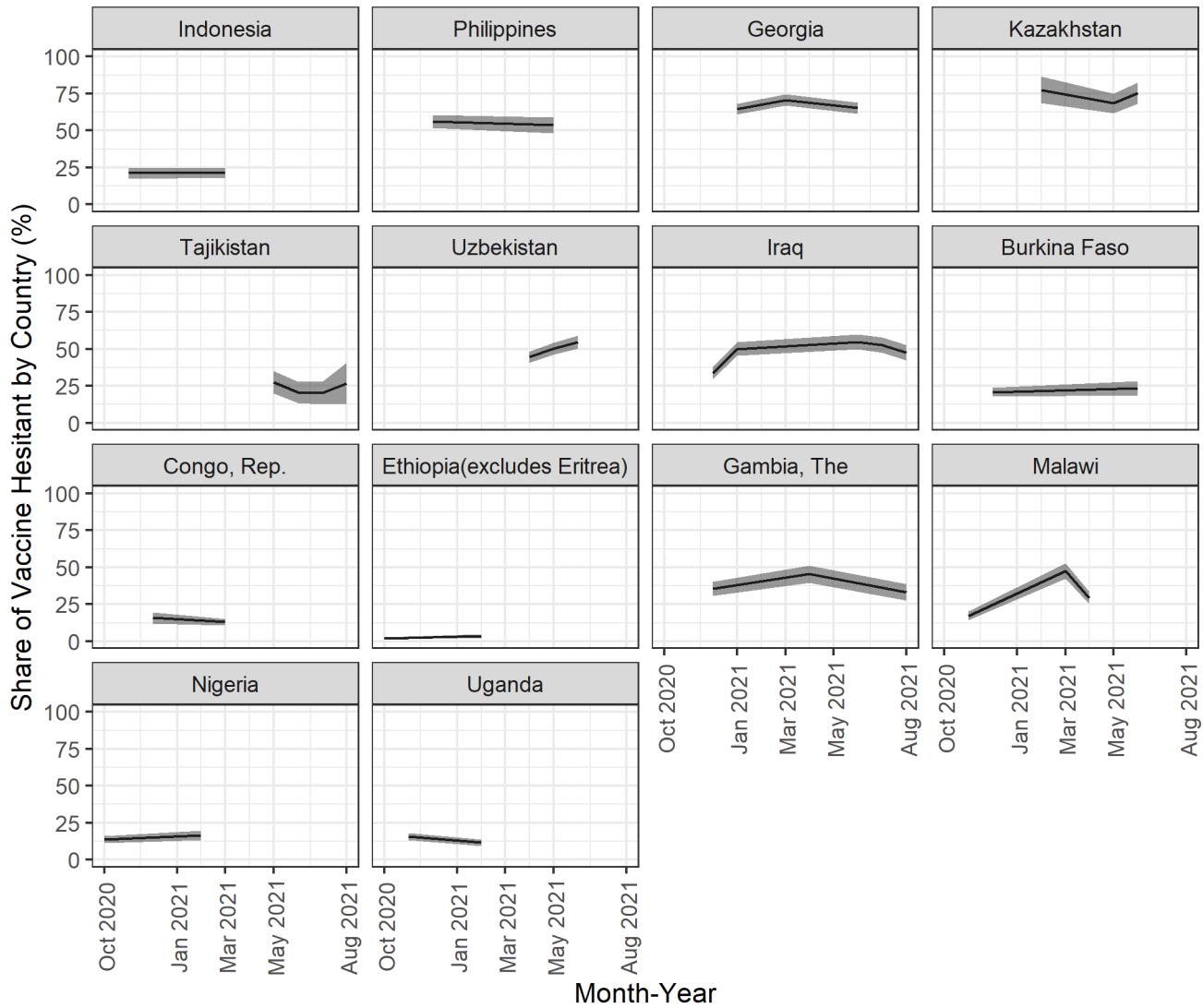

**Figure 3** Changes in levels of vaccine hesitancy in 14 countries, October 2020 to April 2021. The shaded area around the line represents the confidence bands. Source: authors' calculations.

also tested whether trust in the government in general was associated with hesitancy, but this association was not statistically significant (column 5). We also considered a model with vaccine hesitance as an ordinal variable in case the underlying characteristics of those who answered 'no' and 'not sure' differed. The pattern of the results was similar, but the associations were weaker for the 'not sure' group (online supplemental table S5B).

### Changes in vaccine hesitancy over time

In the 14 countries with estimates for two or more survey rounds (figure 3 and online supplemental table S6), there were no clear patterns in terms of the changes in vaccine hesitancy. COVID-19 vaccine hesitancy declined in half of the countries and increased in the other half, and changes in either direction were less than five percentage points in all but three countries (Iraq, Malawi, Uzbekistan). No patterns of change over time were observed across subpopulations either.

### Reasons for COVID-19 vaccine hesitancy

Overall, the most common concern pertained to the safety of the vaccine, including concerns about side effects (43% of respondents in counties with the single answer option) (figure 4). The second most common concern related to efficacy (19%) and each other reason accounted for less than 10% overall. These averages mask significant variation across countries. Concerns about vaccine efficacy were greater or equal to concerns about safety in five Latin American countries (Argentina, Chile, Dominica, Dominican Republic and Mexico). Respondents in the Democratic Republic of Congo and the Republic of Congo were mainly hesitant because of dislike for vaccines in general and distrust for government, pharmaceutical industry and the international community. In Georgia, the only country in Europe and Central Asia with information on reasons for hesitancy, the most cited reason was 'not eligible' (online supplemental figure S4). Only one country, the Republic of Congo, reported

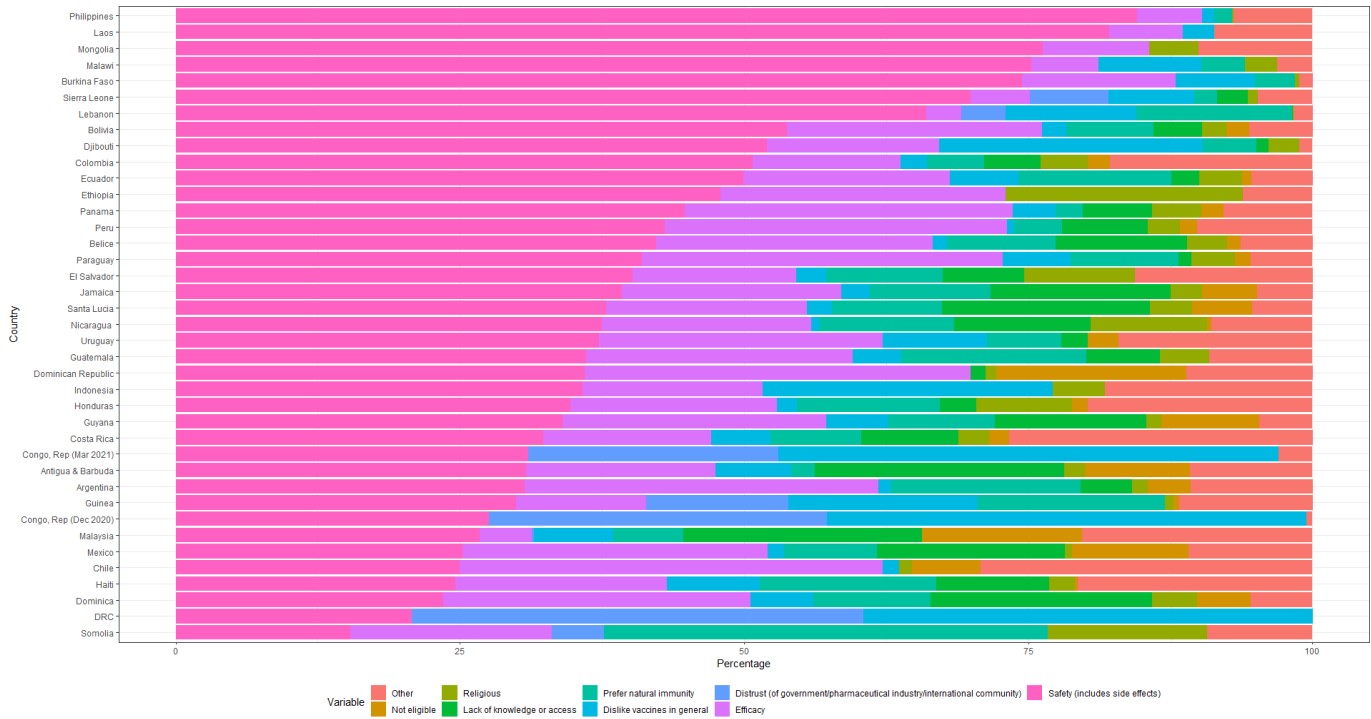

**Figure 4** Self-reported reason for vaccine hesitancy in countries with single answer response, HFPS. Notes: The X axis represents share of vaccine hesitant households. In some cases, the share is larger than 100% because multiple responses were allowed. For the construction of harmonised response categories, see online supplemental table S2B. Results for countries will multiple answer options are shown in online supplemental figure S4. Source: authors' calculations. HFPS, High-Frequency Phone Survey.

reasons for two waves, and only small changes in reasons were observed.

## DISCUSSION

### Principal findings

On average across 53 countries, one in five adults was hesitant about getting a COVID-19 vaccine. The highest levels were observed in Eastern European and Central Asian countries and the lowest levels in Latin American and Caribbean countries. Female respondents, younger adults and those with less formal education reported higher levels of COVID-19 vaccine hesitancy than their respective counterparts. Between October 2020 and August 2021, little change in levels of hesitancy was observed except in Iraq, Malawi and Uzbekistan, where hesitancy increased. The main self-reported reason for being hesitant was concerns over safety, especially worries about side effects.

### Strengths and comparisons with other studies

This study has several advantages over existing literature. It is based on a very large sample of respondents from national surveys in 53 low-income and middle-income countries, a part of the world that is under-represented in the literature. Over half of the surveys used RDD, typically in upper-middle-income countries in which a large share of the population uses mobile phones. In lower-income contexts, most surveys, 35% overall, sampled based on

previous face-to-face surveys which were in turn drawn from a census frame. In the remaining 10% of cases, sampling was carried out from lists provided by mobile phone operators. In each case, surveys drawn from preexisting face-to-face surveys were reweighted using the baseline data to become more representative, while many RDD surveys were also reweighted to make those samples more representative of the universe of phone numbers. While it is impossible to eliminate issues of representativeness in phone surveys, the HFPSs were typically carried out by National Statistics Offices and are more plausibly representative than convenience web surveys.

Overall, the estimates of COVID-19 vaccine hesitancy from the HFPSs were more conservative than other published estimates (online supplemental figure S5). There are several possible explanations for the differences. The first relates to the timing of data collection. Most of the existing literature is based on surveys carried out in mid to late 2020,[14 15 17 19 20] whereas the HFPS data included in our sample were collected from end-2020 to August 2021. It is plausible to believe COVID-19 vaccine hesitancy generally declined during this period as multiple vaccines became available, and widespread vaccination was safely rolled out in high-income countries. Another possible reason for the differences is the variation in how the vaccine hesitancy question was framed and the response options available. For example, some studies used a 4-point or 5-point Likert

scale of agreement,[14–16 20 27] whereas the HFPS relied on two or three answer options. Finally, variations in survey modality may have created biases. Among the published studies included for comparison, many studies were based on data from commercial online sample providers, often using quotas sampling to ensure an appropriate distribution in terms of gender, age and region,[14–17 20 27] while others were based on convenience samples,[11 28–30] or a mix of methods.[19 21] There was also wide variety in the survey mode, including online, computer-assisted telephone and face-to-face surveys.

It is not possible to know the extent to which each of the above-mentioned potential sources of bias affect the comparisons of the findings from the HFPS with the others. For example, it was possible to match the Facebook survey results for the month and year of the HFPS for 20 countries, thus taking away any differences in the timing of the survey. Nevertheless, the HFPS estimates were still lower than those reported by Facebook (online supplemental figure S5D), and this could be due to differences in question-and-answer wording and/or the sample frame.

Even though the HFPS estimates are lower than other sources, the levels of COVID-19 vaccine hesitancy reported here are nonetheless higher than the levels of vaccine hesitancy reported for childhood vaccines. We compared the average country levels of COVID-19 vaccine hesitancy from this study with results from 2019 Global Monitor for 45 countries with observations in both studies.[31] Overall, the levels of COVID-19 vaccine hesitancy reported here are higher than concerns about vaccines in general (online supplemental figure S6). However, there are also similarities in the pattern: countries in Europe and Central Asia and some sub-Sahara African countries reported the highest levels of vaccine hesitancy even before the pandemic, and the biggest reason for pre-COVID-19 vaccine hesitancy was concerns about safety.

Nevertheless, the higher levels of hesitancy reported for the COVID-19 vaccine suggest that respondents are more concerned about the COVID-19 vaccine than about childhood vaccines. The result is even more striking considering that the COVID-19 vaccine hesitancy is estimated rather conservatively in the HFPS, when compared against estimates from other sources. Possible reasons for the higher rates of COVID-19 vaccine hesitancy include the newness of a vaccine that employs an innovative technology (mRNA), it is rapid development and streamlined approval process, and unknown long-term effects. This is consistent with the reasons given by respondents, which are mainly around safety.

Our findings that women, younger adults and those with less education are more vaccine hesitant are largely consistent with results presented elsewhere.[15 19] Although Lazarus *et al* found that men were slightly more hesitant than women, the gender difference was small.[14] The findings with respect to age were also similar.[14 15] The low levels of vaccine hesitancy reported among the older adults provide an important opportunity for vaccine campaigns to target this demographic, especially given recent research that targeting vaccines to older age groups saves the most lives and is highly cost-effective.[32 33] The findings on education are also consistent with other studies,[14 15] although Solís Arce *et al* reported mixed results.[19] Knowing that citizens who are younger and with less formal education are the most likely to be vaccine hesitant can help vaccination campaigns target these population groups. Although we hypothesised that confidence in the government in general would be negatively associated with vaccine hesitancy, this association was not statistically significant in the regression analysis. This is likely because trust indicators were only available at the national level, making the estimates imprecise. Previous studies have shown mixed results,[14 15 18 21] and future analysis is needed to investigate this relationship, preferably using an individual level measure of trust in government.

Our results showed that changes over time were minimal in all countries except Iraq, Malawi and Uzbekistan, where we report relatively large increases. It is difficult to identify the factors that led to increased hesitancy in these cases. The larger body on changes over time in vaccine hesitancy during the same time frame as the HFPS data were collected is mostly from high-income countries and showed mixed results.[34–38] Furthermore, this period was relatively short.

The main reasons reported in the HFPS regarding COVID-19 vaccine hesitancy, which mainly revolve around safety and to a lesser degree efficacy suggest that factors in the 'thinking and feeling' domain of the BeSD framework were predominant.[8] This is consistent with the few studies that report reasons for COVID-19 vaccine hesitancy in low-income and middle-income countries and with the main reasons for vaccine hesitancy historically.[16 19 31]

The variation in COVID-19 vaccine hesitancy across regions, and especially in Eastern Europe and Central Asia where rates are highest, is not fully explained by results in this study. Self-reported reasons for being hesitant were only available for one country in Eastern Europe and Central Asia (Georgia). Although our findings from Georgia that counterindication was a common barrier to vaccination are consistent with results published elsewhere,[39] more research is needed to understand differences in vaccine hesitancy across regions.

Although concerns have been raised about supply constraints being the greatest barrier to COVID-19 vaccination scale-up in low-income and middle-income countries, our study did not find this to be a major concern for most respondents. However, with only one in five individuals in low-income countries vaccinated to date, there was still a large share of the population who would get vaccinated if vaccines were made available to them, and alleviating access constraints will remain a policy priority in such contexts.

## Limitations of the study

The high-frequency surveys used in this analysis were designed to be nationally representative; however, there are potential limitations to phone surveys.[40 41] Phone surveys exclude respondents without access to a phone. This method of data collection was necessary to collect information quickly during the early months of the COVID-19 pandemic while respecting local movement restrictions and minimising the risk of COVID-19 transmission. Given that having a phone may be non-random, this might have created a risk that the results are representative only of the population with access to a phone. However, in the countries in this sample, access to mobile phones was high, and in countries where the sample was based on an existing nationally representative (prepandemic) survey, sampling weights were used in the analysis to correct for the biases resulting from non-random access to phones. Studies using phone survey from Africa also included in our sample have shown that although phone survey respondents tended to be older and better educated, the weighting procedures successfully minimised the selection bias in the phone surveys for a wide range of indicators.[42 43] Another study on labour market participation during COVID-19 using the same phone survey data found that reweighting did not alter the main results.[44]

Another limitation is that the majority of the countries are located in two regions—Latin America and the Caribbean (45% of countries in sample) and sub-Saharan African (26%), making results less representative of other regions.

The uneven roll-out of the COVID-19 vaccine, which occurred earlier in higher-income countries than in lower-income countries, may have affected the reported levels of hesitancy. Most of the survey results reported here were collected after roll-out began in high-income countries but before they were locally available, and this might have led to higher levels of hesitancy than post roll-out. In addition, influential events occurred during the data collection period, including news about adverse side effects of certain vaccines, which could have changed respondents' intentions. The type of vaccine or its country of origin available or anticipated to become available in each country may have influenced levels of hesitancy.

## CONCLUSIONS

Between late 2020 and the first half of 2021, the level of COVID-19 vaccine hesitancy was on average about 20% across 53 low-income and middle-income countries. Although less prevalent than previously thought, COVID-19 vaccine hesitancy was higher than levels of hesitancy reported towards other vaccines, indicating the challenges in scaling up COVID-19 vaccination campaigns may be even greater than for other diseases.

This study provides a rich set of results which can aid policy-makers in the design of vaccination campaigns in low-income and middle-income countries. Overall,

knowing that hesitancy is less prevalent than previous estimates may bolster the case for expanding vaccination interventions. Specifically, knowing that people over the age of 65 years are the least likely to be hesitant provides an important rationale for scaling up vaccine roll-out in this population group, which is at highest risk of severe disease and mortality from COVID-19. In many countries, a sizeable share of the hesitant reported being 'not sure' about getting vaccinating (rather than definitely not willing), likely providing greater opportunity to influence vaccination decisions in these countries. In addition, the findings suggest the importance of designing vaccination campaigns that address concerns about safety, especially about side effects and effectively reach the most hesitant, including women, younger adults and less educated adults.

**Acknowledgements** The authors are grateful to Benu Bidani, Luc Laviolette, Luis-Felipe Lopez-Calva, Carolina Sanchez-Paramo, Juan Pablo Uribe and Monique Vledder for their support of this research. The authors also thank Nobuo Yoshida and the World Bank Data for Goals team for harmonising the high frequency phone survey data and Renos Vakis, Sven Neelsen and the anonymous referees for useful comments on an earlier draft of the paper. The findings, interpretations and conclusions expressed in this paper are entirely those of the authors. They do not necessarily represent the views of the International Bank for Reconstruction and Development/World Bank and its affiliated organizations, or those of the Executive Directors of the World Bank or the governments they represent.

**Contributors** All authors had full access to all the data in the study and take responsibility for the integrity of the data and the accuracy of the data analysis. All authors were responsible for its conception and design. INE, AC and GDB were responsible for the preparation and analysis of the data and JDE, DN, AC, YSK, JM and JJK contributed to their interpretation. JDE drafted the manuscript, and AC, DN, YSK made critical revision of the manuscript for important intellectual content. The corresponding author attests that all listed authors meet authorship criteria and that no others meeting criteria have been omitted. JDE is the guarantor of this study.

**Funding** This paper was funded by the World Bank (Project AA-P177656-GPLG-BB) and the Bill & Melinda Gates Foundation through the Umbrella Facility for Poverty and Equity (TF0B7242), with support from the Global Financing Facility for Women, Children and Adolescents (GFF).

**Competing interests** None declared.

**Patient and public involvement** Patients and/or the public were not involved in the design, or conduct, or reporting or dissemination plans of this research.

**Patient consent for publication** Not applicable.

**Ethics approval** This study involves human participants. Each phone survey was implemented by the respective national statistical office or, in a few cases, a private firm. Informed consent was received from all survey respondents in each country. The World Bank does not require institutional ethics approval for household surveys that are partly or fully financed by the World Bank, including these national phone surveys that inform our research.

**Provenance and peer review** Not commissioned; externally peer reviewed.

**Data availability statement** Data are available in a public, open access repository. Data for most countries are available in a public, open access repository available from the World Bank's Microdata Library, High Frequency Phone Survey Catalog (https://microdata.worldbank.org/index.php/catalog/hfps).

**ORCID iD**
Julia Dayton Eberwein http://orcid.org/0000-0001-8936-6233

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
