## [Reviewer comments · BMJ Open]

ARTICLE DETAILS

TITLE (PROVISIONAL)	How prevalent is COVID-19 Vaccine Hesitancy in Low- and Middle-Income Countries and What Are the Key Drivers of Hesitancy? Results from 53 Countries
AUTHORS	Dayton Eberwein, Julia; Edochie, Ifeanyi; Newhouse, David; Cojocaru, Alexandru; Bopahbe, Gildas; Kakietek, Jakub; Kim, Yeon Soo; Montes, Jose

VERSION 1 – REVIEW

REVIEWER	Patricia O'Malley Premier Health System Support- Nursing Research, Nursing Research
REVIEW RETURNED	11-Nov-2022

GENERAL COMMENTS	Well written analysis. You provided the reader accurate interpretation of the findings as well as detailed multiple factors that may have influenced your results. Supplement 1 description of methods is very clear! The detail of this analysis will certainly provide a basis for further work in reducing vaccine hesitancy. I have a couple of suggestions (minor). 1. Table 1 (page 8), and Supplemental table S5a/b, (page 41,43)perhaps include a legend for the reader to interpret the acronyms.
--

REVIEWER	Maria Pia Fantini University of Bologna, Department of Biomedical and Neuromotor Sciences
REVIEW RETURNED	06-Feb-2023

GENERAL COMMENTS	Reviewer's comments: The paper is of value and suitable for publication after major revision in the introduction and discussion sections. Despite not providing much novelty in terms of evidence for vaccine hesitancy overall, the main strength of the paper is the representativeness of the sample. Specifically, they investigate VH in developing countries which were less studied as compare to high-income ones. The main critique, in my opinion, is the need for a more robust conceptual framework to investigate vaccine hesitancy, referring to recent work by WHO. Please find the following references list which be of help in conceptualizing the introduction and the discussion. Maietti E, Reno C, Sanmarchi F, Montalti M, Fantini MP, Gori D. Are psychological status and trust in information related to vaccine hesitancy during COVID-19 pandemic? A latent class and mediation analyses in Italy. Hum Vaccin Immunother. 2022;18(7):2157622.
--

	doi:10.1080/21645515.2022.2157622 https://www.tandfonline.com/doi/full/10.1080/21645515.2022.2157622
--	---

REVIEWER	Ryoko Sato Harvard University
REVIEW RETURNED	17-Apr-2023

GENERAL COMMENTS	I enjoyed reading the paper. I only have a minor comment.  - Phone survey: representativeness of the sample – can you compare sociodemographic characteristics (education, income, ...) between samples from the phone survey and other nationally representative surveys and see if they are aligned? - Table 1: who is the reference group for the education? Tertiary group? Could you make the lowest education group as the reference group? - It is definitely an interesting study, given the large coverage of countries. However, I also find it difficult to interpret results, especially around the determinants of vaccine hesitancy. - Any thoughts on why there are a substantial variation in the vaccine hesitancy by region? I find it very interesting that central Asia have highest vaccine hesitancy and Latin America has the lowest. Any insights would be helpful.
--

VERSION 1 – AUTHOR RESPONSE

Response to Reviewer 1:

Thank you for your suggestion to include a legend to the tables. We have included the following text to the “Notes” in Table 1 and Supplemental Table S5 a/b: “Ref=reference group; EAP= East Asia and the Pacific; ECA= Europe and Central Asia; LAC=Latin America and the Caribbean; MNA= Middle East and North Africa; SSA= Sub-Saharan Africa; LIC=Low-income countries; LMIC=Lower middle-income countries; UMIC=Upper middle-income countries.”

Response from authors to Reviewer 2:

Thank you very much for the suggestion to include a more robust conceptual framework for investigating vaccine hesitance. We have added discussion of the BeSD conceptual framework (Brewer et al 2017) endorsed by the WHO. See the additional text in the introduction (page 3) and discussion (12).

Response from authors to Reviewer 3:

1. The requested comparison of sociodemographic characteristics between phone surveys and other nationally representative face-to-face surveys was beyond the scope of our analysis. However, such analysis has already been undertaken by others (using the same survey data). (See Ambel et al 2021, Brubaker et al. 2021, and Kugler et al. 2021). Overall, the phone survey samples tend to be older and better-educated than nationally representative face-to-face surveys, but we have been able to use weights to largely correct for this bias this in the countries with existing (pre-pandemic) surveys. We summarize these findings regarding potential biases of phone surveys more fully in the “limitations” section of the discussion.
2. Regarding the education variable, we have changed the reference group to “no education” as requested by the reviewer.
3. Regarding the reasons for variation in vaccine hesitancy by region, we have added the following text to the discussion: “The variation in COVID-19 vaccine hesitancy across regions, and especially in Eastern Europe and Central Asia where rates are highest, is not fully explained by results in this study. Self-reported reasons for being hesitant were only available for one country in Eastern Europe

and Central Asia (Georgia). Although our findings from Georgia that counterindication was a common barrier to vaccination are consistent with results published elsewhere (38), more research is needed to understand differences in vaccine hesitancy across regions.”

VERSION 2 – REVIEW

REVIEWER	Patricia O'Malley Premier Health System Support- Nursing Research, Nursing Research
REVIEW RETURNED	08-Aug-2023

GENERAL COMMENTS	This draft is much more succinct for the reader- well written and organized. References appropriate. Supplemental tables/figures are very helpful in evaluating the author’s reports and conclusions. There are four factors I would ask the authors to consider with regard to this draft. 1. The sample for this study is described frequently as “developing countries”. Considering the list of participating countries provided as well as income, phone availability and other factors, I am not convinced that this term is appropriate to describe the sample. Is the term “developing countries” appropriate? Are developing countries a part of the sample and if so- what part? 2. When the participating country list is examined, 47% of subjects are from South America, 25% from Africa, 19% from Asia, 6% from Europe and 4% from the Middle East. Therefore, this “global study” of 53 countries is in reality a more “local” evaluation of vaccine hesitancy primarily in South America and Africa (72% of the respondents) with attitudes also measured in Asia, Europe and the Middle East which only accounted for 18% of respondents. Should these factors also be addressed briefly in the reporting of results as well as in the study implications? 3. Could the timeline of vaccine release in the countries studied may also be a factor influencing vaccine hesitancy? The African continent began vaccinations in March of 2021 whereas other areas such as Europe and the Middle East began in December 2020. Do the authors believe this historical information would impact their results and conclusions? 4. What statistical software was used for data analysis? In the document, several tables and figures are described as “author calculations”. This helps the reader address the potential for error in data reporting. Note: I reviewed copy with author edits - 2nd draft
---

VERSION 2 – AUTHOR RESPONSE

Reviewer: 1

Dr. Patricia O'Malley, Premier Health System Support- Nursing Research, Indiana University East

Comments to the Author:

This draft is much more succinct for the reader- well written and organized. References appropriate. Supplemental tables/figures are very helpful in evaluating the author’s reports and conclusions.

There are four factors I would ask the authors to consider with regard to this draft.

1. The sample for this study is described frequently as “developing countries”. Considering the list of participating countries provided as well as income, phone availability and other factors, I am not convinced that this term is appropriate to describe the sample. Is the term “developing countries” appropriate? Are developing countries a part of the sample and if so- what part?

Author Response: The authors agree that the term "developing countries" is not clear and have clarified by using the term “low- and middle-income countries” defined according to the World Bank’s country classification.

2. When the participating country list is examined, 47% of subjects are from South America, 25% from Africa, 19% from Asia, 6% from Europe and 4% from the Middle East. Therefore, this “global study” of 53 countries is in reality a more “local” evaluation of vaccine hesitancy primarily in South America and Africa (72% of the respondents) with attitudes also measured in Asia, Europe and the Middle East which only accounted for 18% of respondents. Should these factors also be addressed briefly in the reporting of results as well as in the study implications?

Author Response: Thank you for this suggestion. The authors agree that it is important to note that most of the countries are from Latin America and the Caribbean and Sub-Saharan Africa. We have added the following text to the limitations: “Another limitation is that the majority of the countries are located in two regions -- Latin America and the Caribbean 45% of countries in sample) and Sub-Saharan African (26%) – making the results less representative of countries in other regions.”

3. Could the timeline of vaccine release in the countries studied may also be a factor influencing vaccine hesitancy? The African continent began vaccinations in March of 2021 whereas other areas such as Europe and the Middle East began in December 2020. Do the authors believe this historical information would impact their results and conclusions?

Author Response: Yes, the author agree that the timeline could have affected levels of hesitancy. We have revised the following paragraph in the limitations section accordingly:

“The uneven rollout of the COVID-19 vaccine, which occurred earlier in higher-income countries than in lower-income countries, may have affected the reported levels of hesitancy. Most of the survey results reported here were collected after rollout began in high-income countries but before they were locally available, and this might have led to higher levels of hesitancy than post-rollout.”

4. What statistical software was used for data analysis? In the document, several tables and figures are described as “author calculations”. This helps the reader address the potential for error in data reporting.

Author Response: We have revised the data section manuscript to state that we used R for all the tables and figures and STATA for the regressions.

VERSION 3 – REVIEW

REVIEWER	Patricia O'Malley Premier Health System Support- Nursing Research, Nursing Research
REVIEW RETURNED	26-Sep-2023

GENERAL COMMENTS	This revision is complete and has resulted in a much stronger paper. All the requests to the author(s) related to clarifications of terminology, statistical analysis, and accounting for historical influences on the study results are addressed in this revision. Tables and supplemental materials are clear and support this revised paper as well. This well written paper provides much to reflect upon and will significantly contribute to the literature regarding the impact of the worldwide COVID pandemic.
--